# Small Molecules Targeting INSM1 for the Treatment of High-Risk Neuroblastoma

**DOI:** 10.3390/biology12081134

**Published:** 2023-08-15

**Authors:** Michael S. Lan, Chiachen Chen

**Affiliations:** Department of Genetics, Louisiana State University Health Sciences Center, New Orleans, LA 70112, USA; cchen7@lsuhsc.edu

**Keywords:** INSM1, MYCN, neuroblastoma, neuroendocrine, bioassay, small molecule

## Abstract

**Simple Summary:**

NB is a heterogenous childhood cancer from spontaneous regression to high-risk advanced malignancy. When the tumor is localized, it is generally treatable. However, the long-term survival rate for advanced metastatic NB is poor. Therefore, it is important to further investigate the specific target critical for suppression of NB tumor cell growth as a new treatment protocol that could lead to NB therapy. This review article discusses a recent approach of targeting a new NB biomarker, INSM1 and identifies several potential small molecules that inhibit INSM1 expression and its associated signaling pathway axis in NB. The newly identified small molecules represent either novel or FDA-approved repurposing compounds that could be beneficial for the treatment of high-risk NB.

**Abstract:**

Human neuroblastoma (NB) is the most common childhood extracranial tumor arising from the sympathetic nervous system. It is also a clinically heterogeneous disease that ranges from spontaneous regression to high-risk stage 4 disease. The cause of this disease remains elusive. However, the amplification of *NMYC* oncogene occurred in roughly 30% of NB patients, which strongly correlated with the advanced stage of disease subtype and the worse prognosis status. We discovered that N-Myc oncoprotein binds and activates INSM1, a zinc-finger transcription factor of neuroendocrine tumors. We also found that INSM1 modulates N-Myc stability mediated through PI3K/AKT/GSK3β signaling pathway. Therefore, INSM1 emerges as a critical co-player with N-Myc in facilitating NB tumor cell growth and sustaining the advanced stage of malignancy. Using an *INSM1*-promoter driven luciferase screening-platform, we have recently identified fifteen small molecules that negatively regulate INSM1 expression. Interestingly, the identified small molecules can be divided into four large groups of compounds such as cell signaling inhibitor, DNA/RNA inhibitor, HDAC inhibitor, and cardiac glycoside. These findings support the presence of a unique mechanism associated with INSM1 and N-Myc interplay, which is critical in regulating NB tumor cell growth. We discuss the feasibility of identifying novel or repurposing small molecules targeting INSM1 as a potential treatment option for high-risk NB.

## 1. Introduction

Neuroendocrine (NE) organ, also called diffused NE system, consists of multiple organ sites distributed all over the body. The pituitary gland, the parathyroid glands, and the inner layer of the adrenal gland (adrenal medulla) are almost all made up of NE cells. However, other NE sites contain a much smaller population of NE cells such as thymus, kidneys, breast, liver, prostate, the gastrointestinal endocrine system, the respiratory endocrine system, skin, cervix, ovaries, and testicles. The key function of these endocrine cells produces specific hormones for target tissues in maintaining the body’s metabolism [1]. During the NE cell differentiation, they may share similar biomarkers or display their unique differentiation pathways. There are more than a dozen NE tumors, including retinoblastoma, pituitary tumor, medulloblastoma, neuroblastoma (NB), thyroid medullary carcinoma, prostate cancer, breast cancer, genitourinary carcinoma, Merkel cell carcinoma, insulinoma, NE lung cancer, small cell lung carcinoma, pheochromocytoma, and carcinoid (gastro-intestinal system). The use of markers for the phenotypic characterization of NE cells has yielded a wealth of information on number, type, distribution, and function of normal and tumorous NE cells. Most of these NE tumor markers are cytoskeletal proteins, such as neuron-specific enolase (NSE), chromogranin A, Leu-7, synaptophysin, and NCAM (CD56) [2]. They are valuable when defining the secretory products of the cells and tumors but provide little information beyond their nature and embryological derivation. Each NE tumor type is unique but all the NE tumors as a group share similarity.

To dissect whether there is a common NE target not only specific for NE tumor detection, but also for regulation of NE tumor cell growth would be valuable for the treatment of NE tumors. An *insulinoma associated-1* (*INSM1*) gene is a transiently expressed zinc-finger transcription repressor in developing NE cells that was originally identified in a human insulinoma subtraction library [3]. INSM1 displays a unique promoter-controlled expression in NE tumors [4]. INSM1 expression is highly elevated in NE tumors representing a superior, sensitive, and specific biomarker detected in clinical pathological patient samples [5,6,7]. A normal functional role of INSM1 is critical for the development of chromaffin cells, pituitary endocrine cells, pancreatic islets, lung cancer cells of NE origin, early differentiation of sympathetic neurons, sensory neurons of the dorsal root ganglion, and olfactory neurons [7,8,9,10,11,12]. Our recent studies revealed that INSM1 could be a prominent target in the inhibition of N-Myc oncoprotein driving NB. In the present review article, we discuss the identification of small molecules targeting INSM1 expression in N-Myc activated NB.

### 1.1. Characteristics of NB

NB is derived from neuroblasts of the sympathetic nervous system. It is the most common extracranial tumor in children, which begins in the abdomen, either in an adrenal gland or sympathetic nerve ganglia near the spine in the chest, neck, or in the pelvis. Genetic mutations may account part of the early NB development [13]. From familial NB and genetic predisposition studies in rare cohorts of patients, several germline mutations have been associated with a genetic predisposition to NB, including *ALK* gene mutation, *PHOX2B* gene mutation, and deletion at the *1p36* or *11q14-23* locus [14,15,16]. However, it is rarely related to environmental factors [17]. Diagnosis is based on tumor imaging, urine catecholamine metabolites, and tissue biopsy, or found in a baby ultrasound during pregnancy. NB accounts for about 7% of all childhood cancers. NB can be divided into stages according to the INSS or INRG classification systems. The INSS includes stages 1, 2A, 2B, 3, 4, and 4S. Meanwhile, the INRG system categorizes tumors as very low-, low-, intermediate-, and high-risk. Children in the high-risk group have a 5-year survival rate around 40–50% with a long-term survival rate of only 15% and a relapse rate of 40%. Symptoms of NB are lump in the abdomen, lump in the chest, bone pain, weakness, and bruising around the eyes. When symptoms are apparent, a high percentage of NB are already metastases. In high-risk NB patients, the current treatment protocol includes intensive chemotherapy, surgery, radiotherapy, immunotherapy, and differentiation drugs. 20–50% of high-risk patients do not respond to high dose chemotherapy and are progressive. Relapse is very common after therapy. Therefore, novel and new chemotherapy drugs are urgently needed for high-risk NB treatment.

### 1.2. Identification of Biomarker Critical for Aggressive NB

INSM1 is a transcription factor isolated from an insulinoma subtraction library [3]. In an *Insm1* global ablation mouse model, Insm1 was found to be a critical component in the transcriptional network responsible for sympatho-adrenal (SA) lineage differentiation into chromaffin cells and sympathetic neurons [8]. The observation of early embryonic lethality due to a deficiency in catecholamine synthesis in *Insm1* mutant mice supports Insm1 playing a crucial role in SA lineage differentiation. The *Insm1* mutant mice showed a marked change in terminal differentiation of chromaffin cells and reduced expression of genes whose protein products control catecholamine synthesis and secretion. Catecholamines are essential for mouse fetal development and survival [18,19]. SA lineage cells are derived from neural crest (NC), which give rise to sympathetic neurons and adrenal chromaffin cells. Dys-regulation of *Insm1* in SA differentiation may interrupt the essential trans-regulatory network genes critical for proper SA differentiation into sympathetic neuron, chromaffin cells, and catecholamine synthesis. Similarity between *Insm1* and *Mash1* mutant mice phenotype supports that *Insm1* is the downstream target gene of Mash1. This was also supported by ASCL1 (*Mash1* human homolog), which regulates *INSM1* gene through *E2-box* binding in NE lung cancer cells. Therefore, in the SA lineage transcriptional network, Insm1 is a new player in the control of sympathetic neuron and chromaffin cell differentiation and is located between Mash1/Phox2b and Phox2a/Hand2/Gata3 transcriptional network. Recent studies of N-Myc-activated NB revealed that INSM1 and N-Myc counter-regulate their expression levels through N-Myc, directly activating INSM1 *E2-box*, whereas INSM1 promotes N-Myc stability [20]. High expression of INSM1 and/or N-Myc facilitates NB tumorigenesis. INSM1 and N-Myc express positively in five NB cell lines and a panel of four patient-derived xenograft (PDX) samples (Figure 1). The N-Myc-activated NBs or PDX samples revealed that high expression levels of INSM1 and/or N-Myc were found in association with high-risk NB. Interestingly, we performed an *IC_50_* HHT drug sensitivity test in NB cells revealing that HHT drug treatment sensitivity is proportional to the expression level of INSM1 [21]. Additionally, N-Myc is an oncogenic driver identified as an important genetic biomarker in high-risk NB. N-Myc amplification and/or high expression levels in NB indicates poor prognosis, rapid tumor progression, and patients with high N-Myc expression had poor survival rate. The interplay between INSM1 and N-Myc in high-risk NB promotes malignant aggression of tumor cells. The positive feed-forward-loop of INSM1 activation and enhanced N-Myc stability is essential for aggressive NB tumor growth and oncogenesis (Figure 2). Lowering the INSM1 protein expression could lead to the destabilization of N-Myc protein, which leads to ubiquitination proteasomal degradation pathway.

### 1.3. Development of a Bioassay Targeting INSM1 in NB

Although N-Myc plays a prominent driver role in the process of developing malignant NB, it is generally considered “undruggable” due to the amplification of multiple copies of *MYCN* gene. Current chemotherapy treatment protocol is mostly against intermediate risk of NB includes cyclophosphamide, cisplatin, or carboplatin, vincristine (microtubule inhibitor), doxorubicin (topoisomerase II inhibitor), etoposide (topoisomerase II inhibitor), topotecan (topoisomerase I inhibitor), melphalan, busulfan, and thiotepa (refer to the American Cancer Society). These drugs have anti-neoplastic effect by suppressing the immune system or they function as inhibitors against cellular components such as microtubule or topoisomerase. Recently, a N-Myc co-factor INSM1 was identified to be closely associated with the malignancy of N-Myc-activated high-risk NB. Direct targeting INSM1 expression should be a feasible approach to modulate both INSM1 and N-Myc expression in NB. Specific INSM1 expression is tightly controlled by *INSM1* proximal promoter sequence (−426/+40 bp), which demonstrated specific expressions of reporter gene (*LacZ* or *thymidine kinase*, tk) in either transgenic animal model or cancer gene therapy experiment [4]. Therefore, an *INSM1*-promoter driven luciferase screening platform was established for drug screening. The *INSM1*-proximal promoter (−426/+40 bp) was subcloned into a *pGL4.18-luc2* reporter vector (Figure 3). The constructed recombinant vector was transfected into NB cells and selected under G418. Luciferase activity corresponded positively with *INSM1*-promoter activity. A comprehensive criterion for validation of a cell-based assay was followed using seeding density, solvent compatibility, intra-plate tests, spatial uniformity, inter-plate, and inter-day tests to achieve the robust statistical analysis goal of Z’ factor, S/W, SW values. A drug screening flowchart was designed to validate positive-hit compounds against off-target effects, chemical quench effects, counter screen for reproducibility, NB viability against control cells for specificity, and in vivo NB animal model validation. Two positive-hit compounds, 5′-iodotubercidin (5′-IT) and homoharringtonine (HHT), showed potent inhibition of *INSM1* promoter activity, N-Myc expression, and NB tumor cell growth [20,21]. Successfully, a specific *INSM1*-promoter driven luciferase assay system was established for novel drug screening using INSM1 expression as a prominent target in high-risk NB treatment.

### 1.4. Small Molecules Target INSM1 Expression and NB Tumor Growth

The conventional screening strategy against NB usually directly targets NB cell proliferation. The screening method has a broad range of cytotoxicity but lacks specificity to a designated target. In contrast to the broad inhibition of NB tumor cell growth, we chose INSM1 as a prominent target for the treatment of N-Myc-activated high-risk NBs. Two chemical libraries (147 compounds in an oncology drugs set and 390 compounds in a natural product library) screened against two *INSM1*-promoter driven luciferase assay platforms constructed in BE2-M17 and IMR-32 NB cells were selected. Multiple small molecules were identified specifically to target INSM1 expression and INSM1-associated signaling axis critical for NB cell survival (Table 1). A list of fifteen small molecules derived from *INSM1*-promoter driven luciferase assay is described below in the context of their names, compound nature, and the *IC_50_* viability (against BE2-M17), which could be potentially useful for NB cancer therapy.

5′-Iodotubercidin (5′-IT)−5-IT is a purine derivative used as an adenosine kinase inhibitor. It increases adenosine concentration in the interstitial fluid that induces an intra- and extra-cellular adenosine imbalance. The adenosine imbalance triggers adenosine receptor-3 signaling that decreases cAMP levels, AKT phosphorylation, and enhances GSK3β activity. The 5′-IT treatment also suppresses β-catenin, lymphoid enhancer-binding factor 1, cyclin D1, N-Myc, and INSM1 expression levels, ultimately leading to apoptosis via caspase-3 and p53 activation [20]. 5-IT is able to interact with DNA bases that leads to DNA damage, verified by induction of DNA breaks and nuclear foci positive for γH2AX and TopBP1 activation of Atm and Chk2, S15 phosphorylation, and up-regulation of p53 [22,23,24].

AKT1 inhibitor, A674563−AKT kinase is a central mediator of the signal transduction pathway, which plays a role in cell transformation and tumor progression. AKT is a key regulator for neuronal survival in the developing nervous system. AKT1 belongs to the AKT subfamily of serine/threonine kinases. A674563 is a selective AKT1 inhibitor. The hyper-phosphorylation of AKT1 was found in over 50% of human tumors thus inhibition of AKT1 kinase activities is the potential therapeutic target for cancer treatment [25]. A674563 can be orally administrated into xenograft mouse models for tumor suppression therapy.

Dactinomycin–Dactinomycin, a chromopeptide antibiotic isolated from *Streptomyces* Species, Dactinomycin, also named as actinomycin D, is the most significant member in the chromopeptide antibiotic family. It was the first antibiotic to show anti-cancer properties and has been used in a variety of cancer treatments since 1954, such as Wilms and Ewing tumors, testicular cancer, sarcoma, and choriocarcinoma. It binds to double-stranded DNA and is dose-dependent on inhibition of DNA/RNA synthesis and blocked RNA chain elongation [26].

Plicamycin–Plicamycin, also known as Mithramycin, was isolated from *Streptomyces Plicatus*. The original medical application for Plicamycin was used in treatment of hypercalcemia and hypercalciuria due to its direct inhibitory effect on bone resorption. Plicamycin is an Mg^2+^-dependent antitumor drug. It inhibits DNA/RNA synthesis through binding to the G/C-rich regions of DNA. Mithramycin A is also a DNA/RNA polymerase inhibitor, a DNA-binding transcriptional inhibitor, and is shown to facilitate tumor necrosis factor (TNF)-α and FAS-ligand induced apoptosis. It was used to treat Paget’s disease of bone, testicular cancer, and chronic myeloid leukemia [27].

Daunorubicin-HCl/Idarubicin-HCl–Both of them are anthracycline antibiotics. Anthracycline family includes doxorubicin. Idarubicin, daunorubicin, and epirubicin were first isolated from the natural compound in 1963 as antibiotics [28]. Because their potent anti-tumor efficacy, they have been used in treatment of multiple solid tumors and the hematological cancers including leukemia, lymphoma, uterine, ovary, and breast cancer. General anti-cancer properties of anthracycline include interaction with DNA, such as intercalation, DNA strand breakage, and inhibition of topoisomerase II activity [29].

Mitoxantrone–Mitoxantrone is a synthetic anthracenedine antibiotic developed in 1980. It is an analog of doxorubicin, which is a well-known anti-neoplastic anthracycline antibiotic. It was used in the treatment of adult AML, hormone refractory prostate cancer, and relapsed hepatocellular carcinoma. Mitoxantrone is a DNA reactive agent that intercalates with DNA through hydrogen binding and then induces crosslinks and DNA strand breaks. It is also a potent topoisomerase II inhibitor to disrupt DNA synthesis [30,31].

Romidepsin/Panobinostat/Trichostatin (TSA)/Parthenolide–These are four histone deacetylase inhibitors (HDACis). HDACi can induce morphological reversion of cells transformed with an oncogene to their normal phenotype. Many cancer cells are susceptible to HDACi as in mouse models. These compounds can reduce tumor growth and metastasis in vivo. Acetylation and deacetylation of histones are critical in the regulation of transcription and regulation of epigenesis in eukaryotic cells. Romidepsin was also known as FK228. FK228 was originally identified as an anti-Ras agent, inhibitor of MAPK signaling pathway and then was known as a potent HDACi [32]. Panobinostat, known as LBH-589, is a potent pan-HDACi and shows highly promising results for treatment of several hematologic cancers such as CTCL, Hodgkin lymphoma, and leukemia [33]. TSA is an antifungal antibiotic isolated from a culture broth of *Streptomyces hygroscopicus* and later demonstrated that it is a strong selective class I and II HDACi. TSA is structurally related to suberoylanilide hydroxamic acid (SAHA), which is clinically used to treat the cutaneous T-cell lymphoma (CTCL) [34]. TSA also acts as a protein synthesis inhibitor. Parthenolide, one of the major *Sesquiterpene lactones* (SLs) obtained from the medicinal plant, feverfew. The original clinical application was migraine, arthritis, fever, and stomachache based on its anti-inflammatory effect. Parthenolides consistently display anti-tumor effect in a variety of cancers including gastric cancer, pancreatic cancer, hepatic cancer, lung cancer, ovarian cancer, glioblastoma, and oral cancer. Many mechanisms have been found that are involved in anti-tumorigenesis of parthenolides including inhibit NF-κβ and MAPK activation. NF-κB signaling pathway is constitutively activated in many types of cancer cells, whereas parthenolide targets and reduces NF-κB activity for anti-cancer effects [35,36].

Homoharringtonine (Omacetaxine)–Homoharringtonine (HHT), a plant alkaloid with anti-tumor properties, was identified almost 50 years ago. HHT extracted from *Cephalotaxus fortunei* Hook plants. Omacetaxine, a semi-synthetic form of HHT is a U.S. FDA approved drug used in treatment for CML patients who have resistance to tyrosine kinase inhibitor treatment. The original anti-tumor effect of HHT was identified because HHT directly binds into the A-site cleft of ribosomes, impairs the elongation of the nascent peptide chain, and inhibits protein synthesis such as Bcr-Abl followed by cell death in hematological cancer cells [37]. Studies indicate that HHT executes its anti-neoplastic effects on various cancer cells by regulating multiple pathways. For example, HHT induces apoptosis in CML and AML through downregulation of Mcl-1, which is an anti-apoptotic protein belong to Bcl2 family [38]. HHT was also found to inhibit NB cell growth and induce apoptosis via downregulation of INSM1/N-Myc axis. It destabilizes N-Myc oncoprotein and inhibits the NB tumorigenesis [21].

Lanatoside C/Proscillaridin/Scillaren A–These are cardiac glycosides (CGs). CG is a large family of naturally derived compounds with diversity but share a common structure motif. CGs are categorized into two classes where they are isolated, either from foxglove and oleander (digitalis, digoxin, oleandrin) as cardenolides, or bufadienolide, which is extracted from plants and animals (toads) [39]. Anti-proliferative effects of CGs in cancer cells were observed for years such as in breast cancer. CG was originally used in treatment of cardiac arrhythmia based on its function on inhibition of Na^+^K^+^-ATPase. CGs have anti-tumorigenesis through affects in multiple pathways, which include reducing the cellular membrane potential, an intracellular concentration of K^+^, increasing the intracellular Na^+^ and Ca^++^ to induce apoptosis, like topoisomerase inhibitor, suppressing NF-κB activity, interacting with phospholipid to interfere receptor binding, and modifying N-glycosylation to inhibit cancer cell migration and invasion [39].

### 1.5. Animal Model for NB Therapy

To validate the efficacy of small molecules in treating high-risk NB in vivo, animal models are commonly used in NB therapy. Patient-derived xenografts (PDXs) model shares similar molecular characteristics, NB markers, invasiveness, and malignancy as those which naturally occur in human high-risk NB [40]. However, there arepros and cons to using PDXs as a preclinical NB model, depending on features and sources of NB PDXs. First, athymic nude mice lack functional T cells, NOD-scid mice lack functional T and B cells, while the NOD-scid-gamma (NSG) strain lack functional T, B, and NK cells. They are immune-compromised animals that preclude immune cells involved in tumor therapeutic study. Second, the injection site of subcutaneous or orthotopic into the adrenal gland could make an immense difference for tumor growth microenvironment. Orthotopic tumors retained a more relevant biological phenotype and spontaneous metastases to distant organs [41]. Third, contamination of murine stromal cells could either facilitate the crosstalk or replace it with patient-derived stromal cells, which can restore partial PDX human microenvironment as shown in breast cancer PDXs [42].

The other model we chose is a *tyrosine-hydroxylase (TH)* promoter-activated *N-Myc* transgenic mouse model. This murine NB model activated by transgenic *MYCN* oncogene, which exhibits similar human N-Myc-activated NB in many aspects of tumor formation, including tumor locations, spinal cord involvement, histological presentations, cellular synapses and granules formation, and gains and losses of syntenic regions of chromosomes [43]. The *THp-N-Myc* mouse tumor is strongly positive for the expression of Insm1, which can be a direct target from our selected small molecules. This model will spontaneously develop abdomen and thoracic tumor masses surrounding the spinal cord after 9–13 weeks in 33% of the hemizygous N-Myc-positive animals, whereas homozygous N-Myc-positive animals displayed both increased incidence and decreased latency of tumor formation, approaching 100% at four months [25]. This transgenic mouse line (*THp-N-Myc*) is available from NCI/NIH. Using this transgenic model has an advantage of testing drug treatment in an immune-competent mouse in contrast to the PDX nude mouse model, which is immune-compromised. In the transgenic model or orthotopic injection of PDX tumor cells, it will be difficult to monitor the NB tumor growth in *THp-N-Myc* transgenic or orthotopic injected mouse model except anticipating that NB tumor occurs after four months (homozygotes). Therefore, it is necessary to detect the presence of NB tumor using a biomarker (neuron-specific enolase, NSE) assay [44]. NSE is a specific marker for neurons and peripheral NE cells. Elevated body fluid level of NSE occurs with malignant proliferation and thus can be of value in diagnosis, staging, and treatment of related NB tumor. Raised serum levels of NSE have been found in all stages of *THp-N-Myc* transgenic and xenograft NB tumor-bearing mouse sera, as compared with negative control animals (Figure 4). Additionally, NSE determination in clinical cord blood offers an early postnatal possibility of confirming the diagnosis of NB in newborns [45].

## 2. Discussion

Discovery of new compounds effectively targeting high-risk NB is desirable. To treat high-risk NB patients, we designed a screening platform specific to the INSM1 target that is closely associated with NB oncogenic driver, N-Myc. INSM1 was detected readily in multiple NE tumors and has shown its interactive role with N-Myc contributing to NB malignancy. We have shown that targeting INSM1 could affect N-Myc stability and inhibit NB tumor cell growth [21]. A unique *INSM1*-promoter driven luciferase cell-base assay for chemical library screening was generated. The aims of our study are to explore novel or repurposed small molecules effectively suppressing *INSM1*-promoter activity, which could lead to the down regulation of N-Myc protein and NB tumor cell growth.

Two small molecule libraries obtained from the Drug Synthesis and Chemistry Branch, Developmental Therapeutics Program, Division of Cancer Treatment and Diagnosis, NCI/Bethesda/USA were chosen, due to one of the oncology drug sets (147 compounds) being FDA-approved for various types of cancer treatment. The positive-hit success rate of compounds selected from this library is high (close to 3%), but are specifically against INSM1 expression in NB. These repurposed small molecules have the advantage for NB treatment usage if they can be proven effective in treating NB. The repurposed compound has been approved clinically safe, which could speed up the development and evaluation of its cellular effectiveness and cytotoxicity against NB treatment.

Fifteen small molecules were identified from chemical library screening. Interestingly, multiple families of compounds were found displaying inhibitions of INSM1 expression in NB, such as cell signaling inhibitors, DNA/RNA inhibitors (RNA/topoisomerase inhibitors), HDACis, and cardiac glycosides. Since our assay is designed to target *INSM1* promoter activity, we anticipated the positive-hit compounds would be closely associated with the regulation of *INSM1*-promoter activity (epigenetic modification) and the subsequent interplay with N-Myc protein critical for NB survival. In particular, several compounds including 5′-IT, A674563, and HHT were subjected to an in-depth NB study toward its anti-cancer effects [20,21]. These compounds not only suppressed INSM1 expression readily, but also downregulated N-Myc levels because of N-Myc destabilization. Our approach indicated that suppression of INSM1 expression could be an alternative route of suppressing N-Myc protein to achieve subsequent suppression of NB cells. Different classes of inhibitor may target differential pathways and/or exhibit various cytotoxicity when treated with NB. Potentially, combination of cell signaling inhibitor with DNA/RNA inhibitor, HDACi, or CG could be a promising option for NB treatment. For example, combining HHT and A674563 greatly enhanced inhibition of NB cell viability and cellular proliferation effects (Figure 5) [21]. Combination treatment can lower the dose of various inhibitors, which could achieve a better potency but reduce cellular cytotoxicity, as compared to using single high-dose reagent alone. These unique findings uncover new compounds for the treatment of high-risk NB.

## 3. Conclusions

N-Myc activates INSM1 expression, whereas INSM1 boosts N-Myc stability as a positive loop of NB aggressive malignancy. To identify small molecule compounds targeting INSM1/N-Myc in high-risk NB, we constructed an *INSM1* promoter-driven luciferase assay for screening small molecule library. Fifteen compounds were identified to inhibit *INSM1* promoter activity and NB tumor cell growth. These compounds were inhibitors against cell signaling pathway, DNA/RNA synthesis, HDAC function, and naturally derived cardiac glucoside used in treatment of cardiac arrhythmia. Therefore, our assay is feasible for uncovering novel or repurposed compounds that specifically inhibit INSM1 expression in high-risk NB. INSM1 could be a prominent target for NB therapy.

## Figures and Tables

**Figure 1 biology-12-01134-f001:**
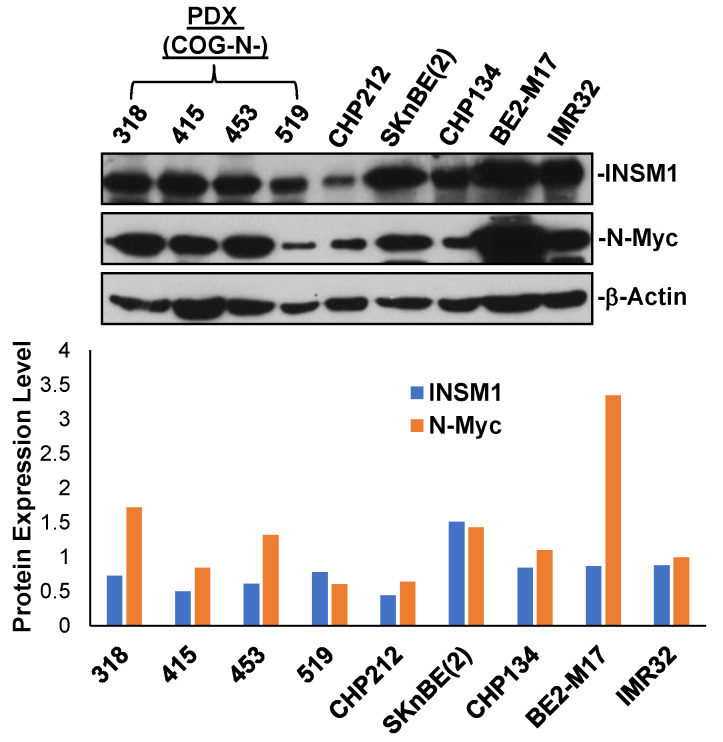
INSM1/N-Myc expression in NB. Five human NB cell lines and four patient-derived xenograft (PDX-COG-N-319, 415, 453, 519) cells were subjected to INSM1 and N-Myc Western blot analysis using β-actin as a loading control. Protein expression level was quantified after β-actin normalization [21].

**Figure 2 biology-12-01134-f002:**
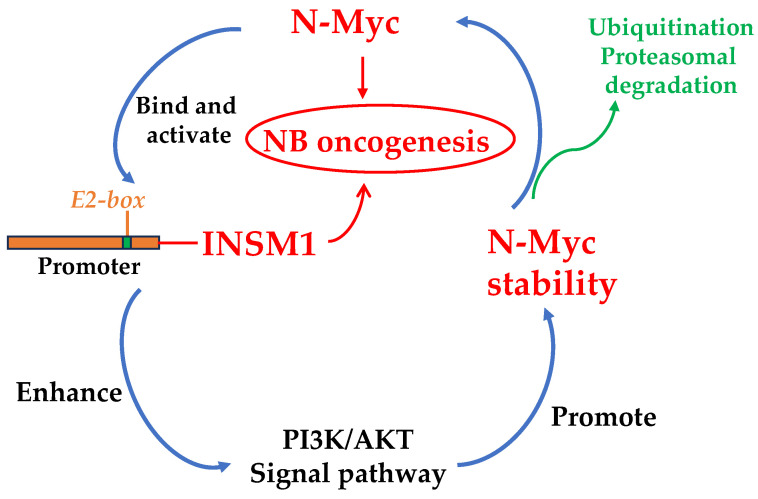
A signaling model describes the INSM1 and N-Myc positive feed-forward loop that facilitates NB cell growth and oncogenesis. N-Myc activates INSM1 expression through E2-box binding, whereas INSM1 expression promotes N-Myc stability, which is critical for NB cell growth.

**Figure 3 biology-12-01134-f003:**
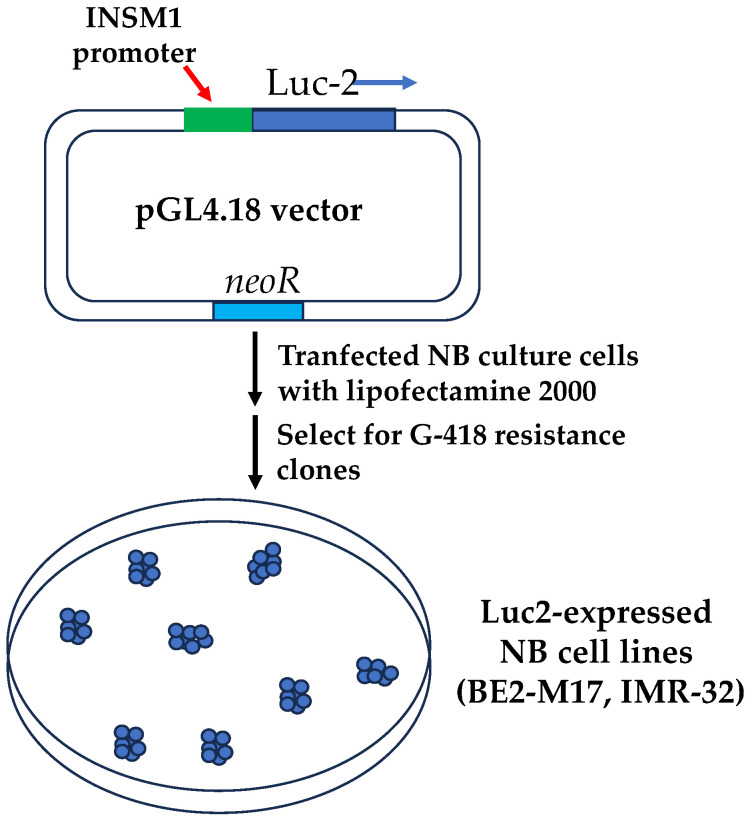
Construction of *INSM1* promoter-driven luciferase reporter into NB cell line. INSM1 proximal promoter (−426/+40 bp) was cloned into pGL4.18-luc2 vector using lipofectamine 2000 and selected with G418. Drug resistant clones express strong luciferase activity.

**Figure 4 biology-12-01134-f004:**
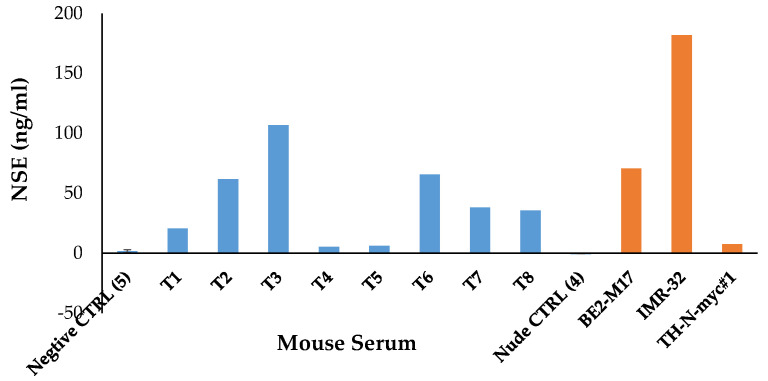
Neuron-specific enolase (NSE) level in NB-bearing mouse. NSE (ng/mL) was measured using ELISA kit in mouse serum from tumor-bearing *THp-N-Myc* transgenic mice (T1-T8) or xenograft NB (BE2-M17, IMR-32, TH-N-myc#1) tumor-bearing nude mouse. Normal mouse sera are used as negative control (CTRL).

**Figure 5 biology-12-01134-f005:**
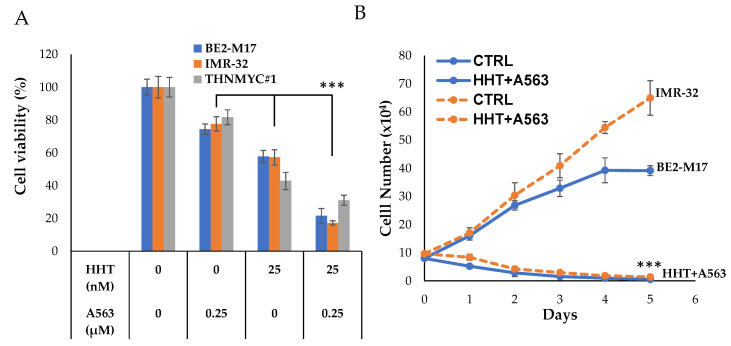
NB combinational therapy using HHT and A563. (**A**) BE2-M17, IMR-32, and TH-N-Myc#1 cell was treated with HHT (25 nM), A563 (0.25 µM), or both for 48 h in a cell viability assay. (**B**) BE2-M17 and IMR-32 cells were treated with HHT (25 nM) + A563 (0.25 µM) for 5 days. Cell numbers were counted using the trypan blue dye exclusion method [21]. ***, *p* < 0.001.

**Table 1 biology-12-01134-t001:** Small molecules inhibit INSM1 expression and NB tumor cell growth.

Name	Compound Nature	*IC_50_*	Ref.
5′-iodotubercidin	Adenosine kinase (ADK) inhibitor.	0.4 µM	[22,23,24]
A674563	AKT1 inhibitor induces apoptosis through p53 and PARP cleavage.	0.28 µM	[25]
Dactinomycin	Also known as actinomycin D preventing RNA elongation, used to treat Wilms tumor, rhabdomyosarcoma, Ewing’s sarcoma, trophoblastic neoplasm, testicular cancer, and certain types of ovarian cancer.	0.03 µM	[26]
Plicamycin	RNA synthesis inhibitor, used to treat testicular cancer, Paget’s disease of bone.	0.062 µM	[27]
Daunorubicin-HCl	Daunorubicin exhibits cytotoxic activity through topoisomerase-mediated interaction with DNA, thereby inhibiting DNA replication, repair, RNA, and protein synthesis, used to treat AML, ALL, CML, and Kaposi’s sarcoma.	0.35 µM	[28,29]
Idarubicin-HCl	It is an anthracycline anti-leukemic drug.	0.68 µM	[28,29]
Mitoxantrone	Topoisomerase II inhibitor, used to treat metastatic breast cancer, acute myeloid leukemia, and non-Hodgkin’s lymphoma.	0.83 µM	[30,31]
Romidepsin	Natural product histone deacetylase inhibitor (HDACi), used in cutaneous T-cell lymphoma (CTCL) and peripheral T-cell lymphomas (PTCLs).	0.2 µM	[32]
Panobinostat	Non-selective pan-histone HDACi (hydroxamic acid), FDA approval for use in patients with multiple myeloma.	0.11 µM	[33]
Trichostatin (TSA)	Class I and II mammalian HDACi family of enzymes inhibit cell cycle progression.	0.086 µM	[34]
Parthenolide	It is a sesquiterpene lactone of the germacranolide class. It inhibits HDAC1 protein, which leads to sustained DNA damage response in certain cells.	0.2 µM	[35,36]
Homoharringtonine (Omacetaxine)	Natural extract, it inhibits protein translation by preventing the initial elongation step of protein synthesis for the treatment of chronic myeloid leukemia (CML).	0.034 µM	[21,37,38]
Lanatoside C	A cardiac glucoside (CG). It was used in complex treatment of patients with heart failure and supraventricular arrhythmias.	0.65 µM	[39]
Proscillaridin	A cardiac glucoside (CG). It was used in the treatment of congestive heart failure and cardiac arrhythmia.	0.013 µM	[39]
Scillaren A	A crystalline steroidal glycoside present in squill or sea onion, which can be hydrolyzed to glucose and proscillaridin A and used as CG.	0.04 µM	[39]

## Data Availability

Not applicable.

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
