# Peer review of "Small Molecules Targeting INSM1 for the Treatment of High-Risk Neuroblastoma"

_biology, 2023, doi:10.3390/biology12081134_

Round 1

Reviewer 1 Report

I think this manuscript is well-written.

In line 81, the authors state 'Cancer can be divided into 6 stages, 1, 2a, 2b, 3, 4, and 4s, which are categorized as low-, intermediate-, and high-risk groups.' However, this seems to mix the INSS and INRG classifications for neuroblastoma. A more accurate representation might be: 'Neuroblastoma can be divided into stages according to the INSS or INRG classification systems. The INSS includes stages 1, 2A, 2B, 3, 4, and 4S. Meanwhile, the INRG system categorizes tumors as very low-, low-, intermediate-, and high-risk.'

Author Response

Response to reviewer #1

There is a mixed statement of the INSS and INRG classifications for NB.

  1. We corrected the statement as “NB can be divided into stages according to the INSS or INRG classification systems. The INSS includes stages 1, 2A, 2B, 3, 4, and 4S. Meanwhile, the INRG system categorizes tumors as very low-, low-, intermediate-, and high-risk”.

Reviewer 2 Report

Lan and Chen in their review entitled “Small Molecules Targeting INSM1 for the Treatment of High-Risk 2 Neuroblastoma” explore the potential of a new biomarker, INSM1, as a therapeutic target in NB and identify several small molecules that inhibit its expression and associated signaling pathways. These small molecules, including cell signaling inhibitors, DNA/RNA inhibitors, HDAC inhibitors, and cardiac glycosides, represent novel or repurposed compounds with potential benefits for high-risk NB treatment. The use of these small molecules to target INSM1 and N-Myc signaling could lead to the development of effective NB therapies.

Neuroblastoma (NB) is a childhood cancer that varies in severity from spontaneous regression to high-risk advanced malignancy. The long-term survival rate for advanced metastatic NB is poor, necessitating the exploration of novel therapeutic targets. This review article discusses the potential of INSM1, a zinc-finger transcription factor of neuroendocrine tumors, as a critical biomarker for NB therapy. Additionally, it identifies small molecules that can inhibit INSM1 expression and its associated signaling pathways, offering new possibilities for high-risk NB treatment. The review identifies fifteen small molecules that specifically target INSM1 expression using an INSM1-promoter driven luciferase screening platform. These small molecules can be categorized into four groups: cell signaling inhibitors, DNA/RNA inhibitors, HDAC inhibitors, and cardiac glycosides. Among them, compounds such as 5'-iodotubercidin (5'-IT), AKT1 inhibitor A674563, and homoharringtonine (HHT) have shown promising effects in inhibiting INSM1 expression and reducing NB tumor cell growth. To validate the efficacy of small molecules in treating high-risk NB, animal models are used, including patient-derived xenografts (PDXs) and a tyrosine-hydroxylase (TH) promoter-activated N-Myc transgenic mouse model. These models provide valuable insights into the potential of small molecules in NB treatment.

Targeting INSM1 represents a promising approach for NB therapy, as it interacts with N-Myc, a key oncogenic driver in high-risk NB. The identified small molecules provide new possibilities for developing effective treatments for high-risk NB. Combining different classes of inhibitors may lead to more effective and tailored therapies for this challenging childhood cancer. Further research and clinical trials are needed to validate these findings and translate them into clinical practice.

I have a major concern with the format of the review. It appears to be confusing as there are experiments performed and included in the manuscript which doesn’t fit the scope of a review. 

A few minor concerns are as follows:

·       Figure 4 lacks error bars. 

·       Figure 5(A) can be rearranged to be more visually clear. 

·       Figure 5(B) has the same color scheme for different entities. 

Author Response

Response to reviewer #2

I have a major concern with the format of the review as experiments performed and included in the manuscript.

  1. In Figure 1, we restated the expression patterns of INSM1 and N-Myc in N-Myc activated NB cells. The reason is to provide a clear visual understanding of INSM1 and N-Myc expression in multiple NB cells. Figure 5 shows the effectiveness of combinational therapy using HHT and A563 to treat NB cell growth. The data was derived from reference 23.

Figure 4 lacks error bars.

  1. Figure 4 shows NSE level in individual tumor-bearing or control mouse (n= 4 or 5). Since serum from each tumor-bearing mouse contains distinct size of NB tumor, their NSE levels vary greatly. Therefore, no error bar was shown. Control mice sera do have error bars, but the NSE are very low close to zero.

Figure 5A can be rearranged to be more visually clear.

  1. We rearranged Figure 5A to make it visually clear.

Figure 5B has the same color scheme for different entities.

  1. We changed the color scheme to make it consistent with the same entity.

Round 2

Reviewer 2 Report

I appreciate the authors' efforts in addressing previous concerns and incorporating revisions. The revisions have significantly strengthened the manuscript, demonstrating the authors' dedication to improving the quality of their work.

Author Response

Thanks for reviewer's positive comment.
